# Role of Inflammatory Biomarkers (NLR, LMR, PLR) in the Prognostication of Malignancy in Indeterminate Thyroid Nodules

**DOI:** 10.3390/ijms24076466

**Published:** 2023-03-30

**Authors:** Claudio Gambardella, Federico Maria Mongardini, Maddalena Paolicelli, Davide Bentivoglio, Giovanni Cozzolino, Roberto Ruggiero, Alessandra Pizza, Salvatore Tolone, Gianmattia del Genio, Simona Parisi, Luigi Brusciano, Loredana Cerbara, Ludovico Docimo, Francesco Saverio Lucido

**Affiliations:** 1Division of General, Oncological, Mini-Invasive and Obesity Surgery, University of Study of Campania “Luigi Vanvitelli”, 80138 Naples, Italy; 2Institute for Research on Population and Social Policies, National Research Council of Italy, 00185 Rome, Italy

**Keywords:** indeterminate thyroid nodules, thyroid cytology, inflammation biomarkers, thyroid cancer

## Abstract

Indeterminate follicular thyroid lesions (Thyr 3A and 3B) account for 10% to 30% of all cytopathologic diagnoses, and their unpredictable behavior represents a hard clinical challenge. The possibility to preoperatively predict malignancy is largely advocated to establish a tailored surgery, preventing diagnostic thyroidectomy. We analyzed the role of the neutrophil-to-lymphocyte ratio (NLR), the platelet-to-lymphocyte ratio (PLR) and the lymphocyte-to-monocyte ratio (LMR) as prognostic factors of malignancy for indeterminate thyroid nodules. In patients affected by cytological Thyr 3A/3B nodules, NLR, PLR and LMR were retrospectively compared and correlated with definitive pathology malignancy, utilizing student’s *t*-test, ROC analysis and logistic regression. One-hundred and thirty-eight patients presented a Thyr 3A and 215 patients presented a Thyr 3B. After the logistic regression, in Thyr 3A, none of the variables were able to predict malignancy. In Thyr 3B, NLR prognosticated thyroid cancer with an AUC value of 0.685 (*p* < 0.0001) and a cut-off of 2.202. The NLR results were also similar when considering the overall cohort. The use of cytological risk stratification in addressing the management of indeterminate thyroid nodules in patients is not always reliable. NLR is an easy and reproducible inflammatory biomarker capable of improving the accuracy of preoperative prognostication of malignancy.

## 1. Introduction

The incidence of papillary thyroid carcinoma (PTC) is rapidly increasing worldwide, due to the increase in differentiated thyroid carcinoma (DTC), probably related to the widespread utilization of high-resolution ultrasound (US) and fine-needle cytology (FNC) [1,2]. Nowadays PTC represents about 94% of all thyroid cancers and 85% of all follicular-derived well-differentiated thyroid malignancies, making it the most frequent endocrine malignancy [3,4,5,6]. With a 10-year survival of approximately 93%, it is considered an indolent tumor. One of the most important targets of the preoperative work-up is the prediction or suspicion of malignancy of a nodule. Neck US is the most adopted method worldwide to stratify the risk of malignancy in thyroid nodules, and a fundamental landmark to indicate further FNC of a suspicious nodule. For the management and follow-up of the PTC, patients are generally classified into low/intermediate- and high-risk groups, according to American Thyroid Association (ATA) Management Guidelines for Adult Patients with Thyroid Nodules and Differentiated Thyroid Cancer [7]. However, clinical and US features are not fully reliable predictors of malignancy, although atypical features and follicular neoplasm cytology are strongly associated with malignancy [8]. In 2014, the major Italian societies involved in the field drafted a new cytological classification to better stratify the pre-surgical risk of thyroid cancer, the SIAPEC-IAP classification, which classifies thyroid nodules into six categories based on FNA results, providing recommendations for their management, and identifying risk factors [9]. The third category includes the indeterminate follicular thyroid lesions (Thyr 3A and Thyr 3B) accounting for 10% to 30% of all cytopathologic diagnoses, and for which the unpredictable behavior represents a hard clinical challenge [9]. Because of the factors related to malignancy, the host immune system may be tightly involved in tumorigenesis, so suppressing the immune response is required to stop the development of the neoplasm. Recently, several systemic inflammatory markers, such as the neutrophil-to-lymphocyte ratio (NLR), the platelet-to-lymphocyte ratio (PLR) and the lymphocyte-to-monocyte ratio (LMR), have been considered predictive and prognostic factors for lung cancer, colorectal cancer, pancreatic cancer, breast cancer and, at least partially, for thyroid cancer [10,11]. In PTC, the predictive value of NLR has not yet been fully investigated. Therefore, the incorporation of the NLR into a prognostic and predictive model might improve the stratification and management of indeterminate risk patients [10,11].

The aim of the current retrospective study is to analyze the role of the NLR, PLR and LMR as prognostic factors of malignancy for indeterminate PTC in Thyr 3A and Thyr 3B patients, comparing their results with definitive pathology in order to stratify the risk and better manage these challenging clinical cases. 

## 2. Results

### 2.1. Study Population

From January 2014 to December 2021, 874 patients were referred to our center for thyroid disease, and 385 presented a preoperative diagnosis of Thyr 3A and Thyr 3B. Only 353 met the inclusion criteria. One hundred and thirty-eight patients presented a cytological diagnosis of Thyr 3A and 215 patients were Thyr 3B. 

Of the overall cohort of 353 patients from our dataset, 247 were female (69.9%) and 108 were male (31.1%), with a mean age of 44.99 ± 14.85 years and a Body Mass Index (BMI) of 25.66 ± 2.83 Kg/m^2^. Demographic and pathological findings are detailed in Table 1.

### 2.2. Study Outcomes

In the Thyr 3A Group, the histological diagnosis was malignant in 14 out of 138 patients (10.1%), while in the Thyr 3B Group, the definitive pathology showed a thyroid cancer in 59 out of 215 patients (27.4%). The histopathological features of the specimens from both groups (size, location and histotype) are summarized in Table 2.

A statistically significant difference was found between WBC and platelet counts in benign and malignant nodules at definitive pathology in the Thyr 3A group, 5666.06 ± 228.03 vs. 4270 ± 1952.34 (*p* = 0.023) and 242,787.6 ± 53,465.55 vs. 214,412.86 ± 36,322.87 (*p* = 0.016), respectively. All the other parameters were not statistically different in Thyr 3A patients [Table 3].

In the Thyr 3B cytological nodules, a statistically significant difference was found between benign and malignant pathology at definitive pathology for neutrophils (3751.19 ± 429.26 vs. 3884.3 ± 416.38, *p* = 0.047), lymphocytes (1848.04 ± 340.47 vs. 1678.64 ± 318.78, *p* = 0.001), LMR (6.33 ± 1.89 vs. 5.56 ± 1.58, *p* = 0.003) and NLR (2.09 ± 0.46 vs. 2.39 ± 0.49, *p* < 0.0001) [Table 4]. All the other parameters were not statistically different [Table 4].

After the logistic regression, in Thyr 3A, none of the variables investigated (LMR, NLR and PLR) were able to predict the malignancy. Only NLR presented an AUC of 0.521 ± 0.079, but the value was not statically significant (*p* = 0.787) with a cut-off according to a max Kolmogorov–Smirnov (KS) of 2.402. Nevertheless, the overall Model Quality of 0.37 was not a good prognostic model, being less than the casualty. 

In Thyr 3B patients, only NLR was successful in predicting thyroid cancer compared to PLR and LMR. The AUC value of LMR was 0.383 ± 0.043 (*p* = 0.006), with a cutoff of 3.398 and a poor overall model quality of 0.28; the AUC of PLR presented a value of 0.541 ± 0.045, but was not statistically significant (*p* = 0.360), with a cut-off of 161.142 and, once again, it had a poor overall model quality of 0.45. Conversely, the AUC value of NLR in Thyr 3B was 0.685 ± 0.041 (*p* < 0.0001), with a cut-off of 2.202 and an overall model quality of 0.60, which is considered a good prognostic model.

Considering the overall cohort (Thyr 3A + Thyr 3B), the only parameter able to significantly predict malignancy was NLR, with an AUC total value of 0.683 ± 0.036 (*p* < 0.0001), a cutoff value of 2.202 and an overall model quality of 0.57. 

In detail, overall, the AUC of PLR was 0.499 ± 0.039, but the value was not statically significant (*p* = 0.988), with a cutoff of 205.599 and a poor overall model quality of 0.42, while the overall AUC of LMR was 0.416 ± 0.038 (*p* = 0.025), with a cutoff of 3.427 and, once again, an overall model quality of 0.34. The results of ROC analysis were depicted in Figure 1, Figure 2 and Figure 3.

The sensitivity, specificity, PPV and NPV associated with the ROC curve analysis of LMR, NLR and PLR were evaluated according to the definitive pathology results in the Thyr 3A and Thyr 3B Groups, and are detailed in Table 5. In detail, only NLR in both groups, in relation to the AUC of the ROC curve, showed a significant specificity of 0.69 in Thyr3A and 0.68 in Thyr3B and a significant negative predictive value of 0.92 and 0.83 in Thyr3A and Thyr3B, respectively [Table 5].

Moreover, hypothesizing that gender, age, subgroup Thyr3A or Thyr3B and the results of NRL, NLR and PLR on blood samples could be related to a causal link to malignancy, a logistic binary model was performed. Thus, belonging to one of the following variables, female gender (*p* < 0.005), NLR (*p* < 0.001) and Thyr 3B (*p* < 0.001), reduced the probability of a benign outcome on histological exam. The female gender, in fact, denoted an increasing risk of malignancy of 1/3 (exp(B) = OR = 0.327). Similarly, high NLR values (>2.202 as cutoff) increased the risk of malignancy of 1/3 (exp(B) = OR = 0.373), while Thyr 3B increased the risk of malignancy of 1/4 (exp/B = OR = 0.248).

## 3. Discussion

This study evaluates the relationship of LMR, NLR and PLR as prognostic factors of malignancy for indeterminate PTC neoplasms in Thyr 3A and Thyr 3B patients, comparing the results with definitive pathology to stratify the risk of malignancy and to better manage these cases. As it is well known, according to international guidelines, indeterminate cytologic PTC management is different; while Thyr 3A can be managed with outpatient control for up to 6 months, Thyr 3B patients often need a surgical approach due to their higher malignancy risk [9,12]. The possibility to prognosticate the malignity of an indeterminate nodule could be of utmost importance since it could prevent a large group of patients from receiving diagnostic surgeries. It is noteworthy that total thyroidectomy and hemithyroidectomy, even in referral centers, are not free from predictable but unpreventable complications [5,6] Nowadays, the key role to address treatment is reserved for the pathologist who cytologically characterizes thyroid nodules after FNC. The cytological examination is the only exam that can provide a definitive preoperative diagnosis with sensitivity and specificity of 68–98% and 56–100%, respectively [9]. 

Subsequently, the evolution of the oncological and immunological research about cancer biology has opened the way for malignancy prediction with immunologic markers [13]. Alterations of LMR, NLR and PLR have been investigated in several tumors of other areas, and seem to have a significant role in PTC progression, but there is still scarce evidence on their malignancy predictive value in Thyr 3A and Thyr 3B patients [3]. Immune cell modification in the peripheral blood, such as thrombocytosis, neutrophilia, lymphocytopenia and monocytopenia, can represent adverse prognostic factors in a wide range of cancers [14]. Neutrophilia can facilitate tumor progression and invasion, while lymphopenia can reduce antineoplastic activities [13]. The role of monocytes and of tumor-associated macrophages (TAMs) has been also investigated in the literature [15,16,17]. They can alter the tumor microenvironment, acting on the growth, angiogenesis, invasion, immunosuppression, chemotherapic resistance and metastasis of different kinds of cancers. In particular, the M1 phenotype of TAM is involved in the inflammatory response and antitumor immunity, whereas the M2 alternative phenotype is related to anti-inflammatory and pro-tumoral progression. Several authors have suggested that the increase in TAMs could be correlated with a worse prognosis [16,17]. Several prognostic scores based on the shifts in these cellular populations have been proposed, including NLR, PLR and LMR. Nevertheless, even benign conditions such as thyroiditis can produce local lymphocytes disorder by increasing thyroid hormones, with a subsequent inflammatory response that alters the normal lymphocyte ratio; therefore, thyroiditis was considered an exclusion criterion in the current study [18]. 

To the best of our knowledge, this is the first study analyzing the predictive value of malignancy in both of the indeterminate thyroid nodules categories, Thyr 3A and Thyr 3B. In the current series, in the Thyr 3A Group, the histological diagnosis was malignant in 14 out of 138 patients (10.1%), while in the Thyr 3B Group, the definitive pathology showed thyroid cancer in 59 out of 215 patients (27.4%), in line with the benchmark of the literature [12,19]. A statistically significant difference was found between only WBC and platelets counts in benign and malignant nodules at definitive pathology of the Thyr 3A group, 5666.06 ± 228.03 vs. 4270 ± 1952.34 (*p* = 0.023) and 242,787.6 ± 53,465.55 vs. 214,412.86 ± 36,322.87 (*p* = 0.016), respectively. In Thyr 3B cytological nodules, a statistically significant difference resulted between benign and malignant tumors at definitive pathology for neutrophils (3751.19 ± 429.26 vs. 3884.3 ± 416.38, *p* = 0.047), lymphocytes (1848.04 ± 340.47 vs. 1678.64 ± 318.78, *p* = 0.001), LMR (6.33 ± 1.89 vs. 5.56 ± 1.58, *p* = 0.003) and NLR (2.09 ± 0.46 vs. 2.39 ± 0.49, *p* < 0.0001).

Considering the ROC analysis, in Thyr 3A, none of the variables investigated (LMR, NLR and PLR) were able to predict the malignancy. Only NLR presented an AUC of 0.521 ± 0.079, but the value was not statically significant (*p* = 0.787). Conversely, in Thyr 3B patients, only the NLR was successful in predicting thyroid cancer compared to PLR and LMR. The AUC value of NLR in Thyr 3B was, in fact, 0.685 ± 0.041 (*p* < 0.0001) with a cut-off of 2.202 and an overall model quality of 0.60, which is considered a good prognostic model [Figure 1, Figure 2 and Figure 3]. Therefore, in the preoperative phase in the case of Thyr 3B cytological diagnosis, a value over 2.202 could be considered highly suggestive of malignancy of thyroid nodules. 

Moreover, only the NLR in both groups, in relation to the AUC of the ROC curve, showed a significant specificity of 0.69 in Thyr3A and 0.68 in Thyr3B, and a significant negative predictive value of 0.92 and 0.83 in Thyr3A and Thyr3B, respectively, confirming the higher propensity of NLR in the detection of healthy subjects.

These conclusions are also supported by the results in the overall cohort of indeterminate cytology (Thyr 3A + Thyr 3B), where the only parameter able to significantly predict malignancy was NLR, with an AUC total value of 0.683 ± 0.036 (*p* < 0.0001), a cutoff value of 2.202 and an overall model quality of 0.57. Additionally, the logistic binary model confirmed that belonging to one of the following variables, female gender (*p* < 0.005), NLR (*p* < 0.001) and Thyr 3B (*p* < 0.001), was associated with a higher malignancy risk.

The results of the current study were encouraging in the higher suspicious category of Thyr 3B nodules, while they were far from being conclusive in the patients affected by Thyr 3A, the higher indeterminate group in which a therapeutical approach is more advocated. 

The role of NLR in malignancy prediction is still a matter of intense debate. Mercier et al. proposed to compare different hematological markers as a prognostic factor for colorectal cancer, showing that a low level of NLR is a prognostic factor for other solid neoplasms, including thyroid cancer [14]. NLR appeared to be the most reliable prognostic factor compared to the other inflammatory markers for head and neck cancers [20]. However, its role in the prediction of thyroid malignancy is still unclear. Seretis’ pilot study proposed an increased NLR value as a prognostic factor for concurrent PTC in preoperative thyroidal goiters and also in incidental PTC [21,22]. Jung et al. reported that a high NLR value (≥1.5) is related to a worse disease-free survival (DFS) rate in stages III-IV PTC patients [23]. Conversely, Ari et al. showed that NLR and PLR can be considered warning markers for PTC or inflammation, but without the ability to discriminate between malignant or benign conditions [24]. Several authors found that LMR was a significant predictor for recurrence-free survival (RFS) in low- to intermediate-risk PTC patients at the time of radio-iodine therapy in high-risk PTC patients [3,4,25]. Offi et al. reported that only LMR was a predictive factor of PTC in indeterminate nodule patients; in fact, high LMR values were associated with cancer. The other parameters analyzed, NLR and PLR, did not show statistical significance as independent PTC prognostic factors [26].

Additionally, PLR was investigated as a predictor of the risk of recurrence and prognosis in patients with PTC [27]. Dolan et al. evidenced the association between elevated NLR and/or PLR and lower LMR with decreased overall survival in patients with advanced inoperable cancer [28,29]. Moreover, the role of inflammatory biomarkers has gained particular interest in the scientific community not only in the oncological field but also in other benign conditions (i.e., coronary artery disease, limb ischemia and diabetes), supporting the relation between proactive inflammatory status and worse outcomes [30,31,32,33].

NLR is a simple biomarker that can be easily determined, and its evaluation is reproducible. These facts make its determination an advantageous tool in the prognostic assessment in association with cytological class, and, in the near future, it could be used in various types of populations to determine thyroid cancer risk, addressing the surgical and medical management of patients affected by indeterminate nodules, especially in the Thyr 3B category.

This study has some limitations to address. Firstly, the exclusion of patients affected by thyroiditis, for its confounding power. Moreover, the limited number of cases reported and the retrospective design of the study are additional limitations.

## 4. Materials and Methods

### 4.1. Study Design

This study is reported according to the STROBE statement for cohort studies [34]. A retrospective analysis was conducted to analyze the malignancy prediction value of the NLR, PLR and LMR in patients affected by preoperative Thyr 3A and 3B nodules, according to Bethesda classification. The results are compared with definitive pathology after thyroid surgery. The study was conducted according to the ethical principles stated in the Declaration of Helsinki. The local institutional ethical committee approved the study protocol. Written informed consent was obtained from all subjects.

### 4.2. Study Setting and Study Population

From January 2014 to December 2021, all patients affected by suspicious thyroid malignancy and referred to the Division of General Surgery of a Teaching Hospital were considered for enrollment in the study. Inclusion criteria were preoperative cytological diagnosis of Thyr 3A or Thyr 3B according to SIAPEC-IAP classification, age ≥ 16 years, American Society of Anesthesiologists (ASA) physical status of grade I or II [12,35,36,37]. Exclusion criteria were presence of highly suspicious thyroid malignancy (Thyr 4 and Thyr 5), presence of preoperative cytology highly suggestive of benign lesions (Thyr 1 and Thyr 2), previous thyroid neoplasms, previous cancer of other areas, coexisting hematologic diseases, additional tumors, acute infectious diseases, chronic drug (steroids, etc.) use that could affect blood analysis, presence of lymphocytic infiltration suggesting thyroiditis on histopathology and abnormal white blood cell (WBC) measurements.

All subjects were preoperatively assessed during a specialized endocrine surgery evaluation. Before surgery, all patients underwent a blood test and serum analysis (including oncological thyroid markers, FT3, FT4, TSH and thyroglobulin), neck ultrasonography, FNC of thyroid nodules, fibro laryngoscopy to evaluate the function of recurrent laryngeal nerve and the glottic space. All patients underwent total thyroidectomy (TT) or hemithyroidectomy (HT) [14]. Clinical data were collected in a prospectively maintained electronic database. The patients were divided into two groups: those with a preoperative diagnosis of Thyr 3A (Group A) and those with a preoperative diagnosis of Thyr 3B (Group B). The preoperative cytological and postoperative histological diagnoses were performed by the institutional Division of Anatomic Pathology.

### 4.3. Predictive Immunologic Markers

Two weeks before surgery, patients from both groups received complete blood count analysis and white blood cell (WBC), neutrophil and lymphocyte counts, obtained using a Dyn Ruby Cell (ABBOTT, Abbott Park, IL, USA) hematology analyzer. In detail, the NLR was calculated by dividing the absolute neutrophil count by the absolute lymphocyte count; similarly, the PLR was calculated by dividing the absolute platelet count by the absolute lymphocyte count; the LMR was obtained by the ratio between absolute lymphocyte count and absolute monocyte count. 

### 4.4. Study Outcomes

The outcome of the study was the evaluation of the ability for malignancy prediction of NLR, PLR and LMR in thyroid indeterminate nodules (Thyr 3A–Group A and Thyr 3B–Group B), analyzing the concordance with postoperative definitive pathology.

### 4.5. Statistical Analysis

Data were described according to each variable type. Continuous variables were expressed as mean with their standard deviation (SD) or median and range. Stata 16 (StataCorp) was used for all statistical analyses. The data were collected on a collective selected through belonging to the Thyr 3A and Thyr 3B categories, and for all subjects included in the analysis, data on malignancy or benignity after histology are available. These parameters identify two main sub-samples: Thyr 3A and Thyr 3B types that could be malignant or benign after histology. The intersection of these parameters also produces the distinction into the four subgroups of interest. For all quantitative variables, or those comparable to the quantitative measurement level, the Student’s *t*-test was used to verify differences in means. The test was performed with the hypothesis of equal variances and with that of different variances. The chosen significance threshold was always *p* < 0.05, meaning confidence in the test outcome greater than 95% is required. For all ordinal or disconnected qualitative variables, the concordance analysis was performed using the Pearson Chi-square test statistic. Additionally, in this case, a significance level of *p* < 0.05 was chosen.

For dichotomous variables (presence or absence of a condition), counts were made in the subgroups without performing probabilistic tests. The variables that were found to be significantly equal in the comparison between subgroups (that is, with the same mean or the same percentage composition in the subgroups) were considered neutral for the purpose of determining the final outcome after histological examination. This indicates that those variables cannot determine differences in the subgroups and therefore were not decisive in pre-determining the conditions favorable or unfavorable to the final outcome.

The analyses were carried out to determine the predictive value of three parameters of interest (described above): LMR, NLR and PLR. Therefore, an ROC analysis was first performed with the aim of measuring the overall diagnostic power of these parameters and that in the Thyr 3A and Thyr 3B subgroups. For the ROC analysis, the sensitivity and specificity curve were compared to the bisector of the first-third quadrant: only a curve that was very high compared to this bisector could be considered diagnostically acceptable. A hypothesis test for evaluating the diagnostic power of each parameter was the one associated with the area under the curve. In this case, too, a value of *p* < 0.05 was considered acceptable. Finally, an overall model quality test was also performed to evaluate the quality of the ROC analysis and it must be greater than 0.5 to ensure good performance. Moreover, sensitivity, specificity, positive predictive value (PPV) and negative predictive value (NPV) of LMR, NLR and PLR were evaluated according to the definitive pathology results in the Thyr 3A and Thyr 3B Groups.

Finally, a binary logistic model was calculated to study the causal relationship between the malignant outcome after histological examination and the values of the predictive parameters. A model with progressive insertion of significant variables was chosen at the level of significance of *p* < 0.05. The β value, OR, i.e., log(B), estimated using the maximum likelihood estimation (MLE) method, represents the risk or protection factor with respect to the malignant outcome after histological examination. When the coefficient B of variable X was positive, then β or OR > 1 was obtained, and therefore Xi corresponded to a risk factor. If the value of βi was negative, then OR < 1, and therefore the variable Xi corresponded to a protective factor.

## 5. Conclusions

The use of cytological risk stratification in addressing the surgical or medical management of patients affected by indeterminate thyroid nodules is not always reliable. The evaluation of an easy and reproducible inflammatory biomarker, the NLR, has proven to improve the accuracy of preoperative prognostication of malignancy, especially in Thyr 3B patients. Therefore, in a near future, this combined cytological and biochemical approach could address a more tailored therapy, preventing diagnostic thyroidectomy. Further, larger comparative studies are necessary to address this issue.

## Figures and Tables

**Figure 1 ijms-24-06466-f001:**
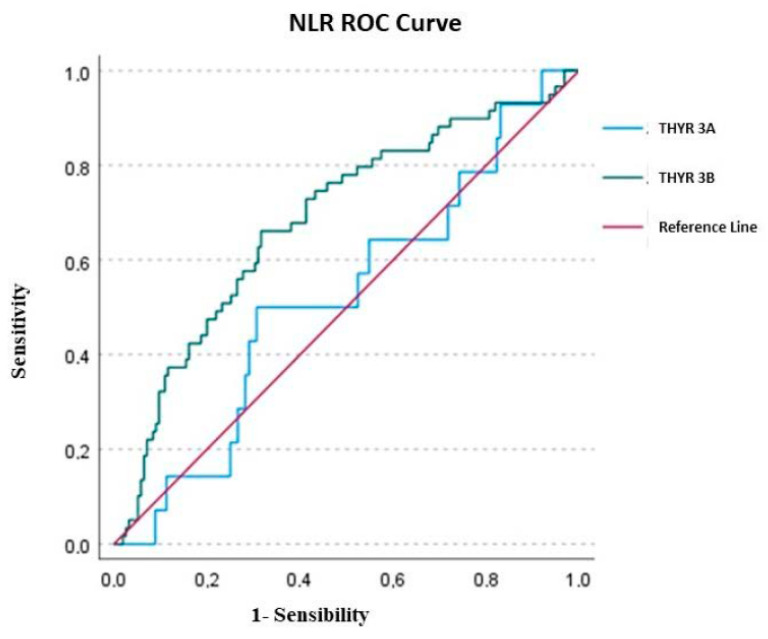
ROC curve of NLR (neutrophil-to-lymphocyte ratio).

**Figure 2 ijms-24-06466-f002:**
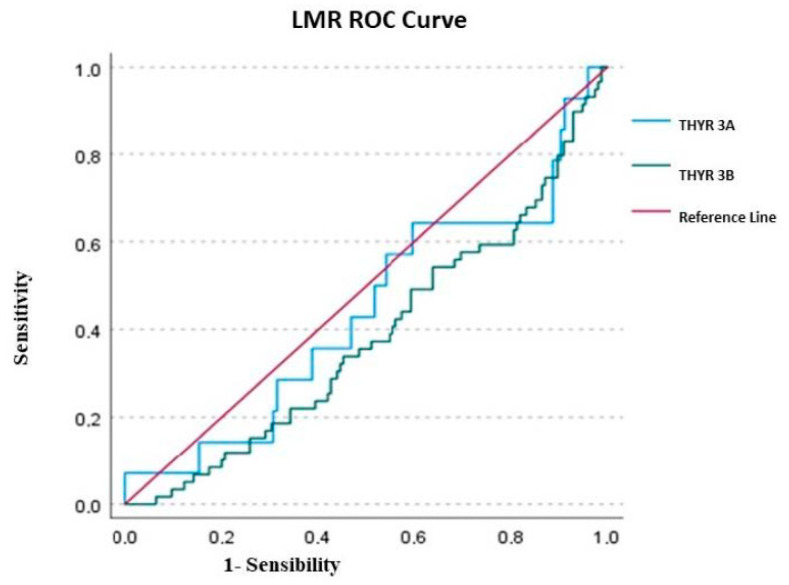
ROC curve of LMR (lymphocyte-to-monocyte ratio).

**Figure 3 ijms-24-06466-f003:**
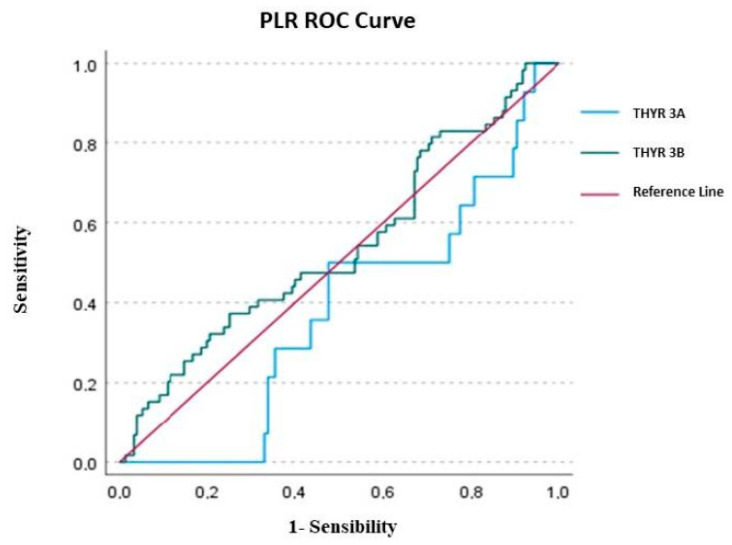
ROC curve of PLR (platelet-to-lymphocyte ratio).

**Table 1 ijms-24-06466-t001:** Demographic and Clinical features of patients affected by THYR 3A and 3B.

	THYR 3A 138 Patients	THYR 3B 215 Patients
**Gender (female/male)**	99/39 (71.7%/28.3%)	148/67 (68.8%/31.2%)
**BMI (kg/m^2^) ***	25.83 ±2.87	26.55 ± 4.81
**Age (years) ***	46.96 ± 14.14	43.73 ± 15.21
**Nodule dimensions (cm)**	105 < 2 17 > 2 and <4 16 > 4	160 < 2 32 > 2 and <4 23 > 4
**Thyroid Lobe**	Right 80 Left 54 Bilateral 4	Right 118 Left 94 Bilateral 3
**Cardiopathy**	8 (5.7%)	27 (12.5%)
**Hypertension**	77 (55.7%)	128 (59.5%)
**Diabetes**	62 (44.9%)	79 (36.7%)
**Renal failure**	9 (6.5%)	8 (3.7%)
**COPD**	35 (25.3%)	36 (16.7%)
**Ischemic Stroke**	12 (8.6%)	6 (2.7%)
**Atrial Fibrillation**	9 (6.5%)	12 (5.6%)
**Dyslipidemia**	22 (15.9%)	31 (14.4%)
**Smoking**	31 (22.5%)	44 (20.5%)

Values are expressed as means ± standard deviation (*) or as raw numbers with percentages in parenthesis. BMI (Body Mass Index), COPD (Chronic Obstructive Pulmonary Disease).

**Table 2 ijms-24-06466-t002:** Overall LMR, NLR, PLR and blood count of patients in THYR3A and 3B Groups.

	THYR 3A 138 Pts	THYR 3B 215 Pz
**White Blood Cell (U/µL)**	5524.43 ± 2288.37	5844.98 ± 2202.76
**Neutrophils (U/µL)**	3746.77 ± 409.41	3787.73 ± 429.80
**Lymphocytes (U/µL)**	1772.77 ± 333.63	1801.55 ± 342.72
**Monocytes (U/µL)**	324.99 ± 63.43	308.78 ± 65.32
**Platelets (U/µL)**	239,909.00 ± 52,588.70	236,924.92 ± 52,119.08
**LMR**	5.69 ± 1.66	6.12 ± 1.83
**NLR**	2.19 ± 0.48	2.18 ± 0.48
**PLR**	141.15 ± 43.61	137.20 ± 43.19

Values are expressed as mean ± standard deviation. NLR (neutrophil-to-lymphocyte ratio), PLR (platelet-to-lymphocyte ratio), LMR (lymphocyte-to-monocyte ratio).

**Table 3 ijms-24-06466-t003:** Comparison of LMR, NLR, PLR and blood count of patients with THYR3A with benign or malignant pathology.

	Benign 124 pts	Malignant 14 pts	*p*
**White Blood Cell (U/µL)**	5666.06 ± 228.03	4270 ± 1952.34	0.023
**Neutrophils (U/µL)**	3735.05 ± 413.29	3850.57 ± 370.67	0.290
**Lymphocytes (U/µL)**	1770.25 ± 334.74	1795.07 ± 335.04	0.796
**Monocytes (U/µL)**	323.03 ± 63.32	342.28 ± 64.07	0.302
**Platelets (U/µL)**	242,787.6 ± 53,465.55	214,412.86 ± 36,322.87	0.016
**LMR**	5.71 ± 1.62	5.53 ± 2.03	0.764
**NLR**	2.19 ± 0.49	2.21 ± 0.43	0.863
**PLR**	143.07 ± 44.28	124.2 ± 33.91	0.072

Values are expressed as mean ± standard deviation. NLR (neutrophil-to-lymphocyte ratio), PLR (platelet-to-lymphocyte ratio), LMR (lymphocyte-to-monocyte ratio).

**Table 4 ijms-24-06466-t004:** Comparison of LMR, NLR, PLR and blood count of patients with THYR3B with benign or malignant pathology.

	Benign 156 pts	Malignant 59 pts	*p*
**White Blood Cell (U/µL)**	5959.06 ± 2169.93	5543.35 ± 2289.12	0.222
**Neutrophils (U/µL)**	3751.19 ± 429.26	3884.3 ± 416.38	0.047
**Lymphocytes (U/µL)**	1848.04 ± 340.47	1678.64 ± 318.78	0.001
**Monocytes (U/µL)**	306.56 ± 66.10	314.64 ± 63.97	0.419
**Platelets (U/µL)**	239,400.35 ± 52,263.05	230,379.69 ± 51,889.76	0.249
**LMR**	6.33 ± 1.89	5.56 ± 1.58	0.003
**NLR**	2.09 ± 0.46	2.39 ± 0.49	<0.0001
**PLR**	134.9 ± 41.91	143.28 ± 46.54	0.227

Values are expressed as mean ± standard deviation. NLR (neutrophil-to-lymphocyte ratio), PLR (platelet-to-lymphocyte ratio), LMR (lymphocyte-to-monocyte ratio).

**Table 5 ijms-24-06466-t005:** True Positive, True Negative, False Positive, False Negative, Positive Predictive Value, Negative Predictive Value, Sensitivity, Specificity of LMR, NLR, PLR in predicting malignant pathology in Thyr3A and Thyr3B patients.

	LMR Thyr3A	LMR Thyr3B	NLR Thyr3A	NLR Thyr3B	PLR Thyr3A	PLR Thyr3B
**True Positive**	13	0	7	39	14	22
**True Negative**	6	2	86	106	7	116
**False Positive**	1	213	38	49	117	39
**False Negative**	118	0	7	21	0	38
**Positive Predictive Value**	0.9286	0.0000	0.1556	0.4432	0.1069	0.3607
**Negative Predictive Value**	0.0484	1.0000	0.9247	0.8346	1.0000	0.7532
**Sensitivity**	0.9286	0.0000	0.1556	0.4432	0.1069	0.3607
**Specificity**	0.8571	0.0093	0.6935	0.6839	0.0565	0.7484

LMR (lymphocyte-to-monocyte ratio), NLR (neutrophil-to-lymphocyte ratio), PLR (platelet-to-lymphocyte ratio).

## Data Availability

The datasets used and/or analyzed during the current study are available from the corresponding author upon reasonable request.

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
