# Peer review of "Role of Inflammatory Biomarkers (NLR, LMR, PLR) in the Prognostication of Malignancy in Indeterminate Thyroid Nodules"

_ijms, 2023, doi:10.3390/ijms24076466_

Round 1

Reviewer 1 Report

The Authors presented an interesting work focusing on the potential prognostic role of inflammatory biomarkers as NLR, PLR, LMR in indeterminate thyroid nodules. 

The paper is well written, methodology and results are clearly reported. 

Even if, results did not highlight that all studied biomarkers could have a potential prognostic role, is important to have data on which perform further studies. 

II would advise the author to briefly implement the introduction/ discussion section adding some comments regarding the great interest that these inflammatory biomarkers are gaining not only in the oncological field but also in other disease reinforcing the background that proactive inflammatory status could be related with worse outcomes. (i.e. cardiovascular disease : doi: 10.3390/biomedicines10092218.; doi: 10.1016/j.atherosclerosis.2012.09.009.; metobil disorders: doi: 10.12669/pjms.336.12900.; surgical outcomes: doi: 10.1177/17085381211010012.; doi: 10.1186/s13104-016-2089-0. etc...)

Minor comments:

-Anamnestic baseline patients features could be implemented (atrial fibrillation, smoking, dyslipidemia, etc...) this data could have an impact on the subsequent analysis due to their potential role on inflammation activation. 

-Please add unity of measures in table II. 

-Hemoglobin level, even if presented in the methods section, is not reported in the analysis 

Author Response

Reviewer 1: The Authors presented an interesting work focusing on the potential prognostic role of inflammatory biomarkers as NLR, PLR, LMR in indeterminate thyroid nodules. The paper is well written, methodology and results are clearly reported. Even if, results did not highlight that all studied biomarkers could have a potential prognostic role, is important to have data on which perform further studies. 

Thank you for you precious comments.

II would advise the author to briefly implement the introduction/ discussion section adding some comments regarding the great interest that these inflammatory biomarkers are gaining not only in the oncological field but also in other disease reinforcing the background that proactive inflammatory status could be related with worse outcomes. (i.e., cardiovascular disease: doi: 10.3390/biomedicines10092218.; doi: 10.1016/j.atherosclerosis.2012.09.009.; metobil disorders: doi: 10.12669/pjms.336.12900.; surgical outcomes: doi: 10.1177/17085381211010012.; doi: 10.1186/s13104-016-2089-0. etc...)

Thanks to the Reviewer’ precious suggestion. The comment and references have been added in the discussion section.

Minor comments:

-Anamnestic baseline patients features could be implemented (atrial fibrillation, smoking, dyslipidaemia, etc...) this data could have an impact on the subsequent analysis due to their potential role on inflammation activation. 

Thanks to the Reviewer’ precious suggestion. The suggested items have been added in the analysis.

-Please add unity of measures in table II. 

The units of measures have been added accordingly.

-Hemoglobin level, even if presented in the methods section, is not reported in the analysis 

Thank you once again for the suggestion. This was mere clerical error and was deleted from methods.

Reviewer 2 Report

The article investigated the prognostic performance of 3 markers in detecting malignancy in indeterminate thyroid nodules.

The article was well presented. Recommend changing the design as a diagnostic testing study:

Obtain TP, TN, FP, and FN values and use e.g MetaDisc to run the analysis. Authors will obtain sensitivity, specificity, likelihood ratios, sROC, ...etc measures to evaluate the diagnostic accuracy of the markers.

Author Response

Reviewer 2: The article investigated the prognostic performance of 3 markers in detecting malignancy in indeterminate thyroid nodules. The article was well presented. Recommend changing the design as a diagnostic testing study: Obtain TP, TN, FP, and FN values and use e.g MetaDisc to run the analysis. Authors will obtain sensitivity, specificity, likelihood ratios, sROC, ...etc measures to evaluate the diagnostic accuracy of the markers.

Thanks for the Reviewer’ precious suggestions and comments. The statistical analyses suggested have been added accordingly in the statistical analysis section and results section.